# Dutch Healthcare Professionals’ Opinion on the Allocation of Responsibilities concerning Prescribing and Administering Medically Indicated Vaccines to Immunocompromised Patients

**DOI:** 10.3390/vaccines11030686

**Published:** 2023-03-17

**Authors:** Elsemieke te Linde, Laura Doornekamp, Katrijn C. P. Daenen, Eric C. M. van Gorp, Anke H. W. Bruns

**Affiliations:** 1Department of Infectious Diseases, University Medical Center Utrecht, 3584 CX Utrecht, The Netherlands; 2Department of Viroscience, Erasmus MC, University Medical Center Rotterdam, 3015 GD Rotterdam, The Netherlands; 3Department of Medical Microbiology and Infectious Diseases, Erasmus MC, University Medical Center Rotterdam, 3015 GD Rotterdam, The Netherlands; 4Department of Infectious Diseases, Erasmus MC, University Medical Center Rotterdam, 3015 GD Rotterdam, The Netherlands

**Keywords:** immunization, vaccination, immunocompromised host, responsibilities, barriers, facilitators

## Abstract

Background: Specific vaccines are indicated for immunocompromised patients (ICPs) due to their vulnerability to infections. Recommendation of these vaccines by healthcare professionals (HCPs) is a crucial facilitator for vaccine uptake. Unfortunately, the responsibilities to recommend and administer these vaccines are not clearly allocated among HCPs involved in the care of adult ICPs. We aimed to evaluate HCPs’ opinions on directorship and their role in facilitating the uptake of medically indicated vaccines as a basis to improve vaccination practices. Methods: A cross-sectional survey was performed among in-hospital medical specialists (MSs), general practitioners (GPs), and public health specialists (PHSs) in the Netherlands to assess their opinion on directorship and the implementation of vaccination care. Additionally, perceived barriers, facilitators, and possible solutions to improve vaccine uptake were investigated. Results: In total, 306 HCPs completed the survey. HCPs almost unanimously (98%) reported that according to them, the primary treating physician is responsible for recommending medically indicated vaccines. Administering these vaccines was seen as a more shared responsibility. The most important barriers experienced by HCPs in recommending and administering were reimbursement problems, a lack of a national vaccination registration system, insufficient collaboration among HCPs, and logistical problems. MSs, GPs and PHSs all mentioned the same three solutions as important strategies to improve vaccination practices, i.e., reimbursement of vaccines, reliable and easily accessible registration of received vaccines, and arrangements for collaboration among the different HCPs that are involved in care. Conclusion: The improvement in vaccination practices in ICPs should focus on better collaboration among MSs, GPs, and PHSs, who should know each other’s expertise; clear agreement on responsibility; reimbursement for vaccines; and the availability of clear registration of vaccination history.

## 1. Introduction

The group of immunocompromised patients (ICPs) is growing, and these patients are at increased risk of infections due to their impaired immune system. Morbidity and mortality in vaccine-preventable diseases, such as herpes zoster or invasive pneumococcal disease, can be reduced by vaccine-induced immunity [1,2]. Vaccine efficacy in ICPs is often found to be only modestly diminished; consequently, recommendations for medically indicated vaccines to ICPs are more often included in international guidelines [3,4,5,6]. However, despite these recommendations, in many countries, including the Netherlands, the uptake of medically indicated vaccines in ICPs is low [7,8,9]. The unawareness of the availability of vaccines and indications for vaccination, distrust in vaccine efficacy, and concerns about the risk of vaccination or fear of deterioration of their underlying disease are important barriers to vaccine uptake in patients [10]. From the healthcare professional (HCP) perspective, lacking clear guidelines and knowledge, time constraints, and reimbursement difficulties are important barriers to discussing or offering medically indicated vaccines [11].

With regard to guidelines, there are more than 18 guidelines on vaccination practices in ICPs involving 21 national and international medical societies and working parties [6,12,13,14]. On top of this, there are several disease-specific recommendations, such as recommendations for stem cell transplantation; solid organ transplantations; hematological malignancies; HIV; and chronic inflammatory diseases such as rheumatoid arthritis, inflammatory bowel diseases, and dermatological conditions. This results in a large heterogeneity in vaccine recommendations, compromising implementation in care [6]. In the Netherlands, a clear vaccination guideline has only been available since 2019 and only for patients with chronic inflammatory diseases [15].

Financial problems from the patient perspective are related to vaccine costs, patients’ inability to pay, and lack of reimbursement, which varies among countries worldwide [11,16,17]. In the Netherlands, vaccines given in hospital or at the municipal public health service are not reimbursed. For specific indications, i.e., for special risk groups, vaccines given by GPs (outpatient medicines) are reimbursed if they are included in the GVS (national medicine reimbursement system). Additionally, there is lack of reimbursement for HCPs providing vaccination care.

These barriers lead to reduced vaccine recommendations by physicians, which is a missed opportunity, since information and recommendation given by HCPs are known as important measures to improve vaccine uptake in ICPs [10,18]. In addition, the clear allocation of responsibilities among HCPs concerning prescribing and administering is often impaired [19]. Consequently, an established vaccination delivery strategy is missing, and ICPs usually miss recommended vaccines [8,9,20]. In the Netherlands, HCPs also face these barriers. Clear agreement on the allocation of responsibilities among HCPs is a prerequisite to overcome the aforementioned barriers to good vaccination practices. We, therefore, aimed to evaluate Dutch HCPs’ opinion on directorship and their role in facilitating the uptake of medically indicated vaccines as a basis to come to an agreement in order to implement and improve vaccination practices in adult ICPs. Additionally, facilitators and barriers for HCPs to recommending and administering medically indicated vaccines and suggestions to improve vaccination practices were measured.

## 2. Materials and Methods

### 2.1. Study Design and Participants

A cross-sectional survey was conducted among in-hospital medical specialists (in training), including nurse practitioners and physician assistants (MSs), general practitioners (GPs), and public health specialists (PHSs), i.e., physicians specialized in infectious disease control, in the Netherlands between February 2022 and July 2022. MSs treating immunocompromised patients aged ≥18 years and working in the field of infectiology, gastro-enterology, rheumatology, hematology, nephrology, ophthalmology, clinical immunology, pulmonology, dermatology, and cardiology were contacted to participate. MSs and PHSs were approached via their professional associations. Only gastro-enterologists with special affinity to inflammatory bowel diseases were contacted. Concerning pulmonology, cardiology, and nephrology, the questionnaire was sent to transplantation pulmonologists, cardiologists, and nephrologists. GPs were approached via the professional network of the authors, as it was not possible to invite them via their professional association.

### 2.2. Data Collection and Analysis

A questionnaire was developed and consisted of three topics: (1) allocation of responsibilities to recommend and administer vaccines; (2) facilitators and perceived barriers to recommending and administering vaccines; and (3) suggestions to improve vaccination practices (the questionnaire is available in the Appendix A). These topics were based on literature studies that reported determinants for vaccine uptake, and national and international reports on vaccination practices in ICPs [10,11,21]. In addition, the baseline characteristics of all respondents were recorded, i.e., age, gender, years of work experience in current profession, and specifically for MSs, their specialization and type of hospital they were affiliated to. The questionnaire consisted of a general part and a second part that was more specifically relevant to the subgroups of HCPs (MSs, GPs, and PHSs) as the relevance and applicability of answer options differed. The questions were close-ended with two different response formats, i.e., a 5-point Likert scale (strongly disagree (1) to strongly agree (5)) and multiple choice. The questions related to all vaccines indicated for ICPs, except for the SARS-CoV2 vaccine and seasonal influenza vaccine, as these are offered programmatically in the Netherlands. The questionnaire was piloted with three MSs and one PHS and amended based on their feedback. We used Castor EDC, an electronic data capture system, to invite the HCPs, and a reminder was sent after two weeks.

### 2.3. Data Analysis

Descriptive analyses were performed on the data. To calculate associations among answers, we used Spearman correlation coefficient for ordinal variables and Phi coefficient for binary variables. *p*-values were considered significant when <0.05.

### 2.4. Ethical Considerations

Informed consent was obtained from all participants with a statement of implicit informed consent included at the beginning of the survey. Participants were informed that participation was voluntary and anonymous. In consultation with the Medical Ethical Research Committee of UMCU, this study was exempted from formal medical ethical review according to the Dutch Medical Research Involving Human Subjects Act [22]. The study complied with Netherlands Code of Conduct for Scientific Practice from the Netherlands Federation of University Medical Centers [23].

## 3. Results

In total, 306 HCPs participated in this study. The largest subpopulation was that of MSs (*n* = 203, including 36 nurse practitioners and 2 physician assistants), followed by 73 PHSs and 30 GPs. A total of 217 (70.9%) were female, and the median age was 44 (IQR 38–52) (Table 1). Among MSs, the four most represented specialisms were infectiology (*n* = 51), gastro-enterology (*n* = 39), rheumatology (*n* = 32), and hematology (*n* = 31) (Table 2). A total of 104 HCPs worked in a non-academic hospital, and 99 HCPs, in an academic hospital. Regarding MSs working in an academic hospital, 88.0% reported that they had the availability of an in-hospital vaccination clinic versus 21.1% of MSs working in a non-academic hospital.

### 3.1. Directorship of Vaccine Recommendations

Almost all HCPs, namely, 98% of MSs, 100% of GPs, and 97% of PHSs, held the primary treating physician responsible for recommending medically indicated vaccines to ICPs. The main responsibility to decide which vaccines are indicated belonged, according to 71% of respondents, to the primary treating physician; according to 65% of respondents, it belonged to a consulted medical specialist, i.e., internist–infectiologist; and according to 32% of respondents, it belonged to a PHS (multiple answers were possible). Noteworthily, PHSs saw themselves (69%) as responsible for deciding which vaccines are indicated, whereas 22% of MSs and 13% of GPs mentioned PHSs as being responsible for it. All three groups mentioned a limited role of GPs in indicating vaccines for ICPs (9%, 3%, and 15% according to MS, GPs, and PHSs, respectively). MSs with assumed more affinity to vaccination (infectiologists, immunologists, and rheumatologists) did not indicate the primary treating physician as being the person responsible for deciding which vaccines are indicated more often than other MSs (Phi coefficient of 0.003 (*p* = 0.973).

### 3.2. Barriers to Recommending Medically Indicated Vaccines

Lack of reimbursement for vaccines, unclear reimbursement arrangements, and lack of insight into patient vaccination history records were the main perceived barriers according to MSs (81%, 80%, and 72%, respectively) and PHSs (81%, 79%, and 48%, respectively) to recommending and actually prescribing the medically indicated vaccines (see Figure 1). GPs also stated insufficient insight into patient vaccination history records and unclear reimbursement arrangements to be important barriers (87% and 70%), as well as ”lack of time” (63%). Almost none of the HCPs considered vaccination not to be effective or not to be indicated (3%).

### 3.3. Responsibility for Vaccine Administration

MSs mentioned various ways to get medically indicated vaccines administered to ICPs. The most mentioned option was to refer the patient to the GP and subsequently a consulted physician in hospital. The primary treating physician and PHSs were equally mentioned (37%). Although the GP was mentioned the most by MSs (62%), only 33% of GPs considered themselves the appropriate professional to administer vaccines. In addition, 37% of MSs mentioned PHSs as an option versus 81% of PHSs themselves. HCPs did not see a clear role for pharmacists in the Netherlands (7% of HCPs).

### 3.4. Barriers and Facilitators to Vaccine Administration

Insufficient reimbursement and lack of clarity about reimbursement were, besides barriers to vaccine recommendation, also barriers to vaccine administration according to MSs (81%). Furthermore, logistic problems (e.g., no nurses available for administering or no vaccines in stock) were mentioned by 61% of MSs. The availability of an in-hospital vaccination clinic was associated with less logistic problems (Spearman correlation coefficient of −0.332 (*p* = 0.000)). GPs mentioned a lack of communication between MSs and GPs, logistic problems, and “not the responsibility of the GP” as the main barriers to vaccination in ICPs (73%, 67%, and 60%, respectively). The lack of reimbursement and insufficient collaboration with MSs were the most important barriers to vaccine administration according to PHSs (69% and 67%, respectively). The lack of experience in the vaccination of ICPs was not a limitation, as almost 90% of PHSs disagreed with that statement.

The most important facilitating factors for MSs to administer vaccines in their medical center were ”convenience for the patient” (69%) and ”assuming that they have the proper expertise” (60%). The former factor was also mentioned by GPs (80%). The majority of GPs did not see “having the proper expertise” and “having most experience with vaccinating” as facilitators. On the contrary, 100% of PHSs mentioned “having a lot of experience” as a facilitator. Other facilitators for PHSs to administer vaccines to ICPs were “adequate knowledge” (99%) and “responsibility of the PHS” (79%).

### 3.5. Solutions to Improve Vaccination Practices

MSs, GPs, and PHSs all mentioned the same three out of nine possible solutions as very important strategies to improve vaccination practices, as indicated by ≥90% of respondents in each group (Figure 2). The first is related to the reimbursement of vaccines. The second important solution for all HCPs is an overview of patient vaccine history. Moreover, the third solution relates to good and clear arrangements on collaboration among the different HCPs. Additionally, MSs indicated the availability of a clear protocol about the practical aspects of administering vaccines to be very important (92%), and GPs indicated the importance of integrated reminders for vaccination in patient files (97%).

## 4. Discussion

In this cross-sectional national survey among more than 300 HCPs, we found that vaccination practices in adult ICPs are suboptimal because of multiple implementation issues in the Netherlands. To the best of our knowledge, this study is the first one that describes the opinions of HCPs at different levels of healthcare, i.e., public healthcare, and primary and secondary/tertiary care, on medically indicated vaccines.

### 4.1. Responsibility to Prescribe and Administer Vaccines

Almost all participating HCPs shared the opinion that the medical specialist, as the primary treating physician, is responsible for recommending medically indicated vaccines to ICPs. Based on this, it seems logical to conclude that the primary treating physician is in charge of recommending vaccines to ICPs. Less agreement existed on which HCP has to decide which vaccines are indicated and who has to administer the vaccines.

Our results clearly show that PHSs saw an important role for themselves in both aspects of vaccination care, although this was not recognized by MSs and GPs. In addition, MSs saw an evident role for GPs in administering vaccines, although GPs did not support that.

These results emphasize the lack of clear agreement and unfamiliarity among groups of HCPs regarding their willingness and ability to contribute to the vaccination care of ICPs. In addition, the data underline the importance of improving recognition and knowledge of colleagues’ vaccination expertise, since nearly 100% of PHSs indicated that they had a lot of experience and knowledge in administering vaccines; however, only 22% of MSs mentioned PHSs as possibly responsible for deciding which vaccines are indicated. A previous—also Dutch—study on strategies to prevent infections in ICPs, not exclusively limited to vaccination, reported that 71% of HCPs thought that the primary treating physician is responsible for administering vaccines in comparison to 37% of MSs in our study [9]. This is probably explained by the fact that the previous study was performed in an academic center; additionally, it was carried out before the COVID-19 pandemic, and the Municipal Public Health Services (MPHS) played a major role in COVID-19 vaccination practices in the Netherlands.

### 4.2. Barriers

In this study, HCPs showed great similarity in their responses concerning factors experienced as barriers to vaccine recommendation, i.e., lack of reimbursement or unclear reimbursement arrangements and unknown vaccination history of patients. GPs additionally mentioned a lack of time. Concerning barriers to administering vaccines, noteworthy is that both GPs and PHSs mentioned insufficient communication and collaboration with MSs.

The perceived barriers for physicians to recommend and prescribe vaccines to ICPs are not extensively studied. However, there are some studies in which barriers for physicians with regard to immunization in subgroups of ICPs were investigated. Two Dutch cross-sectional survey studies investigated perceived barriers for physicians: one examined barriers for nephrologists regarding immunization after renal transplant, and the other, barriers for physicians who deliver care to asplenic patients (internal medicine specialists, surgeons, and GPs) [7,24]. MSs involved in the care of asplenic patients reported poor patient knowledge and a lack of mutual trust between MSs and GPs as barriers. Nephrologists strikingly mentioned that they expected immunization after renal transplantation not to be effective (96.4%), in contrast to our data, according to which only 8% of transplantation doctors (nephrologists, pulmonologists, and cardiologists) indicated to consider vaccinations not to be effective because patients already use immunosuppressives [7].

Our study underscores these barriers previously reported in the literature concerning logistic problems and lack of collaboration among HCPs. Since our study concerned HCPs working in different fields, these barriers are not only related to specific groups of ICPs but to ICPs in general and also exist in a post-COVID-19 setting. A remarkable difference between the referenced study among nephrologists and our study is the opinion on reimbursement. According to our respondents, this is the most important barrier, whereas it was only mentioned by 9% of the nephrologists [7]. A possible explanation for this difference is that most nephrologists did not recommend medically indicated vaccines, such as pneumococcal and tetanus vaccination, and thus did not prescribe them; therefore, they did not encounter reimbursement problems.

Although care systems and collaboration among different specialties vary among countries, the reported barriers in our study are also recognized by non-Dutch physicians. Studies among gastro-enterologists and rheumatologists in the USA and Canada with a similar design also reported a lack of consensus on who is responsible for identifying and administering vaccinations, as well as a lack of knowledge [25,26,27,28]. Other frequently reported barriers are lack of reimbursement, lack of priority, and lack of time [25,29].

### 4.3. Recommendations for Improvements

Not surprisingly, the proposed solutions pertain to the mentioned barriers, and the same three out of nine solutions to improve vaccination practices, i.e., reimbursement of vaccines, clear records of vaccination history, and good arrangements on collaboration among different HCPs, were mentioned by MSs, GPs, and PHSs. Based on the results of this study concerning the allocation of responsibilities, barriers, and proposed solutions, we give the below recommendations.

The first recommendation for the improvement in vaccination in ICPs is rearranging a full reimbursement of medically indicated vaccines independent of the setting and reimbursement for vaccination care provided by HCPs. Because in-hospital vaccinations are not reimbursed, it is a major barrier for the physician to recommend and for the patient to get vaccines.

Secondly, we recommend the clarification of responsibilities among HCPs leading to standardized vaccination responsibilities. Therefore, it is necessary that different HCPs come to an agreement by discussing barriers and concerns. This need is further supported by the finding in our study that the trust in and knowledge of colleagues’ expertise must be improved to strengthen collaboration. This would also make it possible to develop a standardized workflow process in which the roles of different HCPs are clearly described while taking into account the degree of patient complexity. Perhaps, for highly complex ICPs, such as stem cell transplant or solid organ transplant recipients, it is desirable to receive vaccinations in specialized in-hospital vaccination clinics, whereas less complex ICPs, i.e., patients with chronic inflammatory diseases, could be referred to the MPHS for vaccination care. Since PHSs have gained a lot of experience in vaccination care and are willing to play a major role in it, it could be a good alternative to consider and explore this possibility. Additionally, it would reduce the burden on hospital resources.

The third recommendation relates to the importance of clear registration of vaccination history, not limited to the national immunization program for childhood; for example, this could be accomplished by developing a general immunization registry transparent to the involved HCPs or a digital immunization passport held by patients, enabling physicians to offer personalized immunization recommendations [30,31,32]. A previous study among Dutch nephrologists also reported that 82% of nephrologists agreed that the introduction of an immunization passport would improve vaccination care [7]. However, this requires a multidisciplinary approach involving not only HCPs but also policy makers.

Finally, it is well known that a clear protocol about the practical aspects of administering vaccines, as well as electronic reminders for vaccination, is missing, as indicated by 92% and 96% of MSs and GPs, respectively. Vaccine recommendations are scattered over many different guidelines (i.e., programmatic country-specific, vaccine-specific, and illness-specific guidelines exist); information is, therefore, difficult to find. Our last recommendation is to develop a comprehensive manual, based on existing guidelines, on the practical aspects of vaccination care.

### 4.4. Limitations

Although this study comprises all HCPs involved in the vaccination care of ICPs, it has some limitations. Firstly, relatively few GPs were included, as it was not possible to send an invitation via their professional association. This low number has to be taken into account when interpreting the results; as GPs have a central role in the Dutch healthcare system, their widely supported opinion is highly relevant.

Secondly, our study is limited by an uncertain response rate. Only a minority of the addressed representatives of the different professional associations could indicate the number of members who were invited to participate in the study. Therefore, the calculation of response rates per specialism was not possible. Moreover, not all invited physicians treated immunocompromised patients, as it often concerns a subdifferentiation within a specialism, and affinity with and knowledge of vaccination practices may have influenced the response rates. It is plausible that our respondents were chiefly those who had sufficient affinity with vaccination practices to prioritize the questionnaire. Furthermore, to enroll a large population of HCPs in this study, the questionnaire consisted of close-ended questions, which might not have covered all possible answers or completely matched the opinions of HCPs. However, the questionnaire was based on literature studies that reported determinants for vaccine uptake, and national and international reports on vaccine practices in ICPs, and it was developed with experts from two large academic centers in the Netherlands and HCPs in the field of vaccination practices.

## 5. Conclusions

In conclusion, our study demonstrates that the improvement in the vaccination care of ICPs should focus on better collaboration among MSs, GPs, and PHSs, who should be aware of each other’s expertise; clear agreement on responsibility to administer vaccines; and the improvement in logistical aspects, such as reimbursement for vaccines and accessible vaccine registries. This requires a standardized workflow process with clear roles for the involved HCPs and further investigation regarding the implementation of this workflow. Additionally, a sense of urgency in policy bodies is required to improve vaccination practices, since not all barriers can be tracked down solely by physicians.

## Figures and Tables

**Figure 1 vaccines-11-00686-f001:**
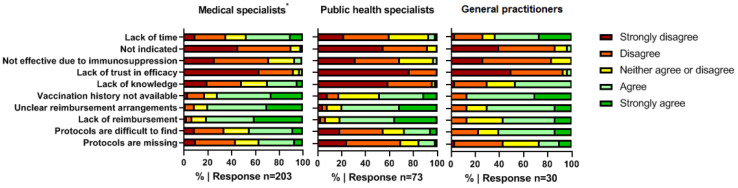
Barriers to indicating medically indicated vaccines. * Including physician assistants and nurse practitioners.

**Figure 2 vaccines-11-00686-f002:**
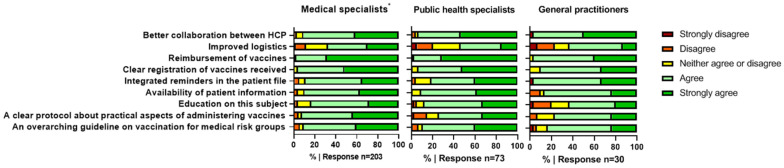
Interventions to overcome barriers to indicating medically indicated vaccines. * Including physician assistants and nurse practitioners.

**Table 1 vaccines-11-00686-t001:** Baseline characteristics of HCPs.

Characteristic	Total	Medical Specialists	Nurse Practitioners/Physician Assistants	Public Health Specialists	General Practitioners
Number (% of total participants)	306 (100)	165 (53.9)	38 (12.4)	73 (23.9)	30 (9.8)
Age (median (IQR))	44.0 (37.8–52.0)	44.0 (39.0–51.0)	40.0 (33.8–48.0)	49.0 (38.0–57.5)	40.0 (34.8–47.0)
Female (*n* (%))	217 (70.9)	107 (64.8)	35 (92.1)	55 (75.3)	20 (66.7)
Years working in current position (%)					
<5	84 (27.5)	46 (27.9)	13 (34.2)	17 (23.3)	8 (26.7)
5–10	74 (24.2)	40 (24.2)	13 (34.2)	11 (15.1)	10 (33.3)
10–15	56 (18.3)	33 (20.0)	7 (18.4)	14 (19.2)	2 (6.7)
15–20	36 (11.8)	16 (9.7)	2 (5.3)	13 (17.8)	5 (16.7)
>20	56 (18.3)	30 (18.2)	3 (7.9)	18 (24.7)	5 (16.7)

IQR: interquartile range.

**Table 2 vaccines-11-00686-t002:** Specialties of medical specialists (including nurse practitioners and physician assistants).

Specialty	MSs (*n* = 203)	
	Non-Academic (*n* = 104)	Academic (*n* = 99)
Infectiology (*n*, %)	29 (27.9)	22 (22.2)
Gastro-enterology (*n*, %)	19 (18.3)	20 (20.2)
Rheumatology (*n*, %)	28 (26.9)	4 (4.0)
Hematology (*n*, %)	20 (19.2)	11 (11.1)
Nephrology (*n*, %)	2 (1.9)	16 (16.2)
Ophthalmology (*n*, %)	0 (0)	8 (8.1)
Clinical immunology (*n*, %)	1 (1.0)	6 (6.1)
Pulmonology (*n*, %)	1 (1.0)	6 (6.1)
Dermatology (*n*, %)	0 (0)	4 (4.0)
Cardiology (*n*, %)	0 (0)	1 (1.0)
Unknown (*n*, %)	4 (3.8)	1 (1.0)

## Data Availability

The data presented in this study are available upon request from the corresponding author. The data are not publicly available due to privacy reasons.

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
