# Peer review of "Dutch Healthcare Professionals’ Opinion on the Allocation of Responsibilities concerning Prescribing and Administering Medically Indicated Vaccines to Immunocompromised Patients"

_vaccines, 2023, doi:10.3390/vaccines11030686_

Round 1

Reviewer 1 Report

The study is of localized interest given that the focus is on Netherland-specific concerns. There is information in the literature for other nations/regions, but because the interest is in developing solutions for the Dutch system in particular, these data are not necessarily relevant, except in seeing whether there are deep systemic differences arising in Holland (there do not seem to be).

The sample is thoughtfully constructed, though a bit small, especially when the team seeks to speak of relative results among subgroups. But the means of seeking respondents is standard as is the means of statistical analysis. It is a straightforward study, professionally executed.

The results and conclusions are communicated clearly and the conclusions well-supported by the data. There is little in this study that is shocking with the possible exception of the difference in opinion concerning vaccination responsibility between those lower down on the health care status ladder when compared with those toward the top. It is a bit surprising that the suggestion is not made that this matter be made a part of the educational/licensure process was not considered despite the fact that it seems a common sense place to standardize understanding.

The only other comparable study, as the piece mentions, was performed before the COVID-19 pandemic, which makes the comparison problematic. As such, it seems important that this study establish a starting point for similar public health research.

The piece will need a good copy editing before publication. There is inconsistent comma usage around "i.e.", there is a sudden increase in font size, a blank section, and a couple of spelling errors.

Reviewer 2 Report

The topic is important, more is needed on HCPs’ views on such matters.  I would suggest expanding the introduction to provide more information on what is already known on this topic and where some of the gaps in knowledge are. For example, rather than saying there may be a lack of clear guidelines, describe briefly the major sets of guidelines developed by international groups and national bodies, as well as distinct specialties which may have specific needs issues regarding some vaccines for their disease states. If there are reimbursement difficulties, where are they—there are multiple facets involved., so what has the literature identified about this as problematic and where are the gaps in knowledge, especially with regard to HCPs and IC patients.

The Methods section: please give details about survey development, if available.  Why was the survey not based on additional sources other than literature and how did you survey that and on what basis did you develop your questions? Please describe the survey questions. Also are there grounds for thinking that your questions have validity? 

What was the sampling universe? Who was this sent to and how? How many hospitals and other settings? And any non-hospitals? What is the rationale for mixing 11 different specialties (as per table 2), who may all have different interests. Why not focus on just academic/tertiary/teaching hospital setting or all at other secondary-level hospitals?  What is the rationale for mixing MS, NPs, PAs, public health workers, and GPs, who are rather different provider types?  Why did you not focus on just one group, or two groups to draw some valid comparisons in an overall sample size of 306? 

There is no indication of how representative this survey may have been. We do not learn what the survey response rate may have been. Please supply these important pieces of information.  Was there any statistical power calculation involved to answer another type of research question?

How do your findings vary by vaccine type for different types of IC patients?  And are you talking about adults of all ages, or including children as well, or which age groups?

The section 4.3, recommendations. for improvements, is rather sweeping because it is drawing potential policy recommendations based on one study for which some details are not presented.  I would recommend making this less bold and putting it in a broader discussion that also seeks to integrate more with other studies published on HCPs’ views about this topic for IC patients.

Reviewer 3 Report

The study aimed to evaluate opinions of Dutch health care professionals concerning their role in recommending and administering vaccines to immunocompromised patients, and additionally to investigate barriers, facilitators, solutions to improve vaccination coverage. The issue is very important since vaccine preventable disease are causes of morbidity and mortality among this growing group of patients. The study is well designed, methods are clearly described and results are appropriately presented. The authors compare their results with other similar studies and try to explain differences. Their conclusions are justified. Additionally, they provide recommendations how to improve vaccines uptake. The main limitation of the study is small number of respondents, especially general practitioners, but this is underlined by the authors.

I have only minor remarks:

1. I would suggest to provide some information in the Introduction concerning vaccination among immunocompromised patients in the Netherlands (if possible): e.g. vaccination coverage against pneumococci, influenza, COVID-19. It would demonstrate how important is to undertake actions to improve.

2. Are any vaccines reimbursed (at least partially) for immunocompromised persons? Is there any difference between pediatric and adult patients? I presume that the study concerns vaccinations of adults; it should mentioned.
